# A qualitative study exploring motivations for participating in research among women who use opioids

Sarah M. Bagley[1,2,3]*, Ariel D. Maschke[4], Miriam T. H. Harris[1,3], Alexander Y. Walley[1,3], Samantha Johnson[1,3], Emily Hurstak[1,3], John Farley[5], Sarah G. Keller[1], Vanessa M. McMahan[5], Cynthia Barrett[5], Phillip O. Coffin[5], Christine M. Gunn[6]

1 Department of Medicine, Boston University School of Medicine, Boston, Massachusetts, United States of America, 2 Department of Pediatrics, Boston University School of Medicine, Boston, Massachusetts, United States of America, 3 Grayken Center for Addiction, Boston Medical Center, Boston, Massachusetts, United States of America, 4 Crown Family School of Social Work, Policy, and Practice, University of Chicago, Chicago, Illinois, United States of America, 5 San Francisco Department of Public Health. San Francisco, California, United States of America, 6 Dartmouth Institute for Health Policy and Clinical Practice, Geisel School of Medicine Dartmouth College, Lebanon, New Hampshire, United States of America

* sarah.bagley@bmc.org

## Abstract

### Objectives

We assessed the acceptability of four recruitment strategies and explored facilitators and barriers to research engagement among women who use opioids.

### Methods

We recruited self-identified women reporting past 14-day non-prescribed opioid use using four recruitment approaches: community outreach in collaboration with community-partners; snowball sampling; social media campaigns; and passive recruitment through distribution of print materials at community programs. We collected participant demographics, type of recruitment, and substance use via an interview-administered survey. Qualitative interviews explored women's research experiences, and facilitators and barriers to research engagement. Analysis employed a combination of inductive and deductive approaches to identify themes relevant to women's engagement in research.

### Results

Of 36 enrolled participants, median age was 49 years, 16% were Black, 58% were white, 14% were Hispanic, and 58% had their own house or apartment. We recruited 12 women through community outreach, two through snowball sampling, three through social media, and 19 through print materials. Interviews identified four

**Data availability statement:** All relevant data are within the manuscript.

**Funding:** This work was supported by the National Institutes of Health [R01DA045690].

**Competing interests:** The authors have declared that no competing interests exist.

themes: (1) highest trust when recruited through community organizations and lowest trust when recruited through social media; (2) desire to improve the lives of other women who use drugs drove motivation to participate in research, (3) preference for monetary compensation, which increased the likelihood of in research participation; and (4) women participated when research environments were supportive and destigmatized.

## Conclusion

To recruit women who use drugs, researchers should collaborate with trusted community organizations, promote the benefits of research to other women, monetarily and fairly compensate participants, and foster supportive destigmatized environments.

## Introduction

In the United States, among women ages 30–64, the drug overdose death rate increased 260% and the rate of opioid-involved deaths increased 492% from 1999–2017 [1] Though women have lower opioid-related mortality rates relative to men, critical encounter touchpoints (e.g., opioid detoxification or nonfatal opioid overdose) are associated with higher opioid overdose mortality among women, highlighting missed opportunities for overdose prevention for women who use opioids [2] Furthermore, women have unique drivers of opioid-related harms compared to men. For example, benzodiazepines more commonly contribute to fatal opioid overdoses among women and women are twice as likely to be prescribed benzodiazepines in the 30 days prior to an opioid-related death [3]. Women also face greater barriers to accessing harm reduction services and opioid use disorder (OUD) treatment because of violence, stigma, sexual and reproductive health matters, and childcare responsibilities compared to men [4–10]. Women who do receive OUD treatment have fewer years of prior opioid use but more severe psychiatric and medical complications than men [11] Historically, drug use research disproportionately captures the experiences of men despite increasing drug-related risks for women [12].

Intersectional marginalization, the impact of being part of multiple intersecting social categorizations (e.g., race, gender, and persons who uses drugs) that face discrimination and oppression, impacts the differential opioid overdose and harm reduction service access rates among women. A long history of discriminatory drug policies and racism affecting marginalized populations, compounded by the unique challenges faced by women who use drugs, contribute to the higher overdose rates among minoritized women. From 2015 to 2020, the rate of fatal drug overdose rose 144% among Black women, and in 2020 American Indian/Native American women had the highest fatal overdose rate among women, with 32 deaths for every 100,000 individuals [13].

Given the differences in women's experiences with OUD and overdose, there is a need for gender-specific research to develop and test interventions to reduce the

harms related to opioids among women [9]. As the drug supply has become increasingly unreliable due to fentanyl, it is critical that we understand how these changes differentially impact women. In a qualitative study that described concerns that superseded fear of overdose among people who use fentanyl, for example, men were more concerned about violence and incarceration, whereas women reported concern about risks related to their health, safety, and relationship with child protective service [10]. Such findings emphasize the importance of adequately including women in substance use research so that interventions are responsive to women's specific needs in the context of a dynamic drug supply.

Despite the need for increased representation of women who use drugs in addiction services and research, gender-responsive treatment and equitable representation in research remain elusive goals [14]. Studies often have inadequate sample sizes to conduct gender-specific analyses, research protocols for substance use treatment do not usually include potential gender-related mediators or moderators, and women are not adequately represented in research [12]. A recent review of randomized controlled trials registered in clinicaltrials.gov between 2010 and 2019 found only 8% reported sex-specific analyses, and 1.5% reported the inclusion of transgender women [15]. Further, a qualitative study of women who use drugs found that women had "dehumanizing" experiences in research and felt researchers held stigmatizing and inaccurate perceptions of them [16]. Thus there is a need, not only to increase women's representation in research, but to ensure research experiences, protocols, and outcomes are aligned with women's needs and emphasize the value of women's contributions to research. Recruiting women who use opioids into research is a key first step to conducting valid research that can address these needs of women.

We sought both to assess the acceptability of four distinct recruitment strategies and to explore facilitators and barriers to research engagement among a sample of women who use opioids.

## Methods

### Design and setting

We deployed four recruitment strategies to engage self-identified women with past 14-day non-prescribed opioid use in Boston, Massachusetts and San Francisco, California in a qualitative study involving interviews about their perspectives on and experiences with opioid use, substance use research, and recruitment for research studies. This research was reviewed and approved by the Boston Medical Center Institutional Review Board (H-39303) and the University of California, San Francisco Institutional Review Board (19–29181). It was deemed to be exempt research. We provided a research information sheet with the required elements for informed study participation. We asked all participants to agree to the audio recording before we started.

### Recruitment

The study included three planned recruitment strategies at the outset: community outreach in collaboration with community partners, social media, respondent driven snowball sampling [17,18]. In prior studies, we had challenges recruiting young adult women; and thus intended to recruit 15 individuals under age 30 to ensure representation of younger women in this study. We planned to continue enrollment efforts until we reached thematic saturation. Recruitment began on January 11, 2020, and, approximately two months after study initiation, the emergence of COVID-19 restricted in-person recruitment efforts. In response, research staff at both sites implemented a fourth recruitment strategy, passive recruitment, which included distributing study print materials at venues serving women, social service agencies, and in public locations. Recruitment ended on October 5, 2020 when we reached thematic saturation.

**Community outreach in collaboration with community partners.** In Boston, we identified a community-based partner that served people who use drugs. We supported 0.20 FTE of an outreach worker's salary for project training, review of recruitment materials, and to support recruitment efforts at this site. In San Francisco, we provided part-time support to a dedicated recruitment manager who had extensive community research recruitment expertise. The recruiters

approached individuals at community sites to describe the study and assess interest. For interested individuals, recruiters administered screening questions to confirm eligibility and schedule an interview. Eligible participants included English-speaking adults who self-identified as women and reported past 14-day non-prescribed opioid use.

**Social media.** The research team worked with the Boston Medical Center (BMC) marketing team and community harm reduction partners to create social media recruitment materials. We prioritized content that showed women of diverse racial identities and/or an explicit connection to opioid use via traditional advertising formats and memes. In Boston, these materials were posted on the institution's Facebook and Instagram accounts. The BMC marketing team managed two, six-week social media campaigns daily by posting, optimizing, and collecting data. In addition, the community partner posted recruitment materials on their Facebook page. Finally, research staff posted materials via Craigslist ads. In San Francisco, the research team posted recruitment materials to Instagram, Facebook, and Craigslist. San Francisco community-based partners also posted and shared study media on their social media platforms.

**Snowball sampling.** Across the two sites, we aimed to identify and train six recruitment 'seeds' (i.e., study participants who would recruit potential participants through their social networks). To promote participation across diverse groups, we sought to identify seed participants who had enrolled via different recruitment strategies, represented a range of ages (older and younger than 30 years old), and identified as members of different racial or ethnic groups. Participants who completed an interview and met criteria to be a seed were invited to participate in the snowball sampling with two training options. Option 1 was a snowball sampling orientation (10–15 minutes) that covered project goals, study requirements and procedures, eligibility for study participation, and seed recruitment responsibilities. No additional compensation was provided for Option 1. Option 2 (30-minutes) included the Option 1 activities, plus skills-building training, including mock recruitment pitches with the research staff. Option 2 provided $50 in compensation. After completing training, seeds were given three coupons to distribute. When a participant was enrolled as the result of a coupon, the seed whose coupon was linked to the recruitment received an additional $20.

**Passive recruitment.** In both Boston and San Francisco, passive recruitment involved dropping off flyers at organizations that work with people experiencing homelessness, youth, transgender individuals, and people who use drugs.

## Data collection

We collected age, race and ethnicity, highest level of education achieved, housing status, and past 30-day drugs used for each participant. We also collected recruitment process data to describe the number of participants recruited from each strategy, to summarize the number of contacts prior to enrollment, and to record the number of days from initial contact to enrollment. For the BMC social media campaign, we collected the number of impressions (views of the ads as individuals scroll through the feed), post engagements (actions such as "like" or sharing), and link clicks (clicking the study website link).

We conducted qualitative interviews using open-ended, flexible interview guides developed by our study team. Interviews were approximately 45–60 minutes and could be conducted in-person or on the phone by research staff (ADM, VMM, and CB, all identifying as women and with prior experience conducting qualitative interviews). Researchers conducting the interviews did not previously have relationships with the participants and participants did not know the researchers' personal motivations for conducting this study. Interview guides covered: experiences of recruitment for the current study; experiences in other research studies, including motivations for participation, facilitators and barriers to involvement, reimbursement preferences, and recommendations for future research; and experiences with social services and health care related to being a woman and using drugs. There were no other people present during the interviews besides the researchers and the participants. This analysis specifically focused on the first two domains that were related to research participation. All interviews were audio-recorded and transcribed verbatim, reviewed for accuracy, de-identified, and imported into NVivo qualitative data management software. All participants were compensated $40 for completing an interview.

## Analysis

We used descriptive statistics to summarize participant demographics, the proportion of participants recruited by the different recruitment strategies, the number of days from first contact to enrollment, the number of contacts prior to enrollment, and social media interaction data. The study team (AM, SGK, and CMG) drafted a codebook with deductive codes based on the study's goals and the interview guide, which explored women's experiences and perspectives on involvement in research studies. During codebook development, additional inductive codes representing emergent concepts were identified, defined, and applied in subsequent coding. Each transcript was independently coded by two members of the study team (SMB, ADM, SGK, CMG) and discrepancies were resolved in a consensus process including at all members of the coding team. The core coding team met weekly to discuss emerging themes, consensus, and when coding was completed, the interpretation of results. The team included individuals with diverse experience and expertise in community engagement, risk communication, addiction medicine and harm reduction, and women's health. In weekly meetings, the team held robust conversations about data to mitigate potential researcher bias.

Using a qualitative directed content analysis approach, coding queries were used to build themes related to previous research experiences and recommendations for improving women's research participation [19].

## Results

### Recruitment results

Thirty-six women were recruited, 16 in Boston and 20 in San Francisco. Six participants were under the age of 30 years. Of the 36 participants, median age was 49 years; 16% were Black, 58% were white, 14% were Hispanic, 6% Native American/Alaskan Native; 81% had at least a high school degree or equivalent. Past 30-day drug use prevalence by substance included 78% heroin, 75% fentanyl, 75% benzodiazepines, and 56% cocaine (Table 1).

**Sources of recruitment.** Table 2 includes the number of participants who were recruited by each of the four strategies, as well as the mean and range of the number of contacts and days between initial contact and participation. Community outreach yielded 14 participants, 8 in Boston and 6 in San Francisco, only one of whom was under 30 years old. Social media efforts yielded three participants, two of whom were under 30 years old, from Boston and none from San Francisco. Seventeen participants were recruited through passive recruitment with print materials.

Nine participants trained for snowball sampling, six in Boston and three in San Francisco. Three seeds were recruited through community outreach, two by social media, one by another seed, and three through passive recruitment. All nine completed the full training. One seed in Boston recruited two additional participants. The other eight seeds did not recruit additional participants.

**Social media recruitment.** There were 278,504 views of the traditional image ads and 271,277 of the meme ads. Of those, 0.4% (n = 2013) clicked the link to the study website and three participants were recruited. A detailed summary of Boston social media campaigns can be found in Table 3.

### Qualitative thematic analysis

Qualitative analysis identified four themes related to personal and contextual factors that facilitated or impeded participation in substance-related research. Each participant is identified by recruitment method, city, and age category (<30 years old or ≥ 30 years old). Themes and representative quotes are also included in Table 4.

Theme 1: Participants reported high trust when recruited through community organizations and low trust when recruited through social media.

The majority of participants expressed that recruitment through street outreach and existing relationships were the best approaches to engage women who use opioids in research. One woman shared,

**Table 1. Participant demographics and characteristics.**

| | Boston (N = 16) | San Francisco (N = 20) | Total (N = 36) |
|---|---|---|---|
| **Recruitment type** | | | |
| Community Partner | 8 (50%) | 6 (30%) | 14 (39%) |
| Snowball sampling | 2 (12%) | 0 | 2 (6%) |
| Social Media | 3 (19%) | 0 | 3 (8%) |
| Passive | 3 (19%) | 14 (70%) | 17 (47%) |
| **Age in years (median, IQR)** | | | |
| | 39 (29.5, 49.75) | 49 (40.25, 52.75) | 49 (34.5, 52) |
| **Race** | | | |
| White | 13 (81%) | 8 (40%) | 21 (58%) |
| Black or African American | 0 | 6 (30%) | 6 (16%) |
| Native American/ Alaskan Native | 0 | 2 (10%) | 2 (6%) |
| Another race | 0 | 2 (10%) | 2 (6%) |
| More than one race | 3 (19%) | 2 (10%) | 5 (14%) |
| **Ethnicity** | | | |
| Hispanic | 1 (7%) | 4 (20%) | 5 (14%) |
| Not Hispanic | 15 (93%) | 16 (80%) | 31 (86%) |
| **Highest level of education achieved** | | | |
| Less than high school | 2 (13%) | 5 (25%) | 7 (19%) |
| High school degree or equivalent | 11 (69%) | 12 (60%) | 23 (63%) |
| Associate's degree | 2 (13%) | 0 | 2 (6%) |
| Bachelor's degree | 1 (6%) | 1 (5%) | 2 (6%) |
| Other | 0 | 2 (10%) | 2 (6%) |
| **Past 30-day housing** | | | |
| My own house/ Apartment | 9 (56%) | 12 (60%) | 21 (58%) |
| Someone else's house apartment | 3 (19%) | 0 | 3 (8%) |
| Shelter/On the streets | 3 (19%) | 2 (15%) | 5 (14%) |
| Drug treatment center | 1 (6%) | 0 | 1 (3%) |
| Rented room | 0 | 3 (15%) | 3 (8%) |
| Vehicle | 0 | 1 (5%) | 1 (3%) |
| Other | 0 | 2 (10%) | 2 (6%) |
| **Past 30-day drug use (yes)** | | | |
| Heroin | 13 (81%) | 15 (75%) | 28 (78%) |
| Fentanyl | 14 (88%) | 13 (65%) | 27 (75%) |
| Other opioids/Painkillers | 7 (44%) | 7 (35%) | 14 (39%) |
| Benzodiazepines | 13 (81%) | 14 (70%) | 27 (75%) |
| Cocaine | 8 (50%) | 12 (60%) | 20 (56%) |
| Methamphetamine/Amphetamine | 5 (31%) | 13 (65%) | 18 (50%) |

"…because how can you reach somebody you don't know? To be honest with you, I think that was a good intervention…That would be a good way, if you know somebody. Addicts know addicts. Addicts know that you're not the police." (Community partner, San Francisco, age ≥30)

**Table 2. Recruitment strategies, number of contacts, and number of days from initial contact to confirmation of eligibility and enrollment among women who participated in interviews (n = 36).**

|  | Community partner n = 14 | Snowball sampling n = 2 | Social media n = 3 | Passive recruitment n = 17 |
|---|---|---|---|---|
| Number of days from eligibility confirmed* to enrollment** (mean, range) | 5 (0 - 14) | 3 (1–5) | 6.3 (0 −12) | 3.2 (0–21) |
| Number of contacts with study staff prior to enrollment* (mean, range) | 1 (1–1) n = 13+ | 3.5 (3 - 4) | 4.3 (3 −5) | 1.3 (1–4) |
| Number of days from first contact to enrollment (mean, range) | 7 (0 - 30) | 138 (6–270) | 7.6 (4–12) | 3.6 (0–21) |

*Eligibility confirmed = day that potential participant completed the brief screening agreement and the screener to confirm eligibility.

**Enrollment = day that participant completed the HEAR Exempt Informed Consent procedure and the HEAR interview.

+ Missing 1 data point from San Francisco.

**Table 3. Boston Facebook and Instagram social media Ad campaign results.**

| Result | Traditional Image Ads N (%) | Meme Ads N (%) | Total N (%) |
|---|---|---|---|
| Total number of views | 278,504 | 271,227 | 549,731 |
| Any click, like, or share of a material | 918 (0.3%) | 1,037 (0.4%) | 1,955 (0.4%) |
| Number of times shared | 4 (0.001%) | 34 (0.01%) | 38 (0.007%) |
| Number of links to the study website | 850 (0.3%) | 1,163 (0.4%) | 2,013 (0.4%) |

The majority of links to the study site were from used on a Facebook mobile device (89%, 1,793/2,013).

Women appreciated that community health workers would know them well enough to refer them to relevant studies:

"I mean it feels good that [community health worker] knows me enough to know that I would fit the criteria, and not guessing…That she...thought about me when she was recruiting or whatever, however she does it. She was like, "I thought about you and I thought about this one [study]." (Community partner, Boston, age ≥30)

Women perceived community outreach worker recruitment as decreasing the risk of being turned away for not meeting study criteria. One woman described past frustrations when attempting to participate in opioid-related research:

"Disappointing, sometimes, because they say they want somebody that's on opioids and they want drug users, want injectors and I go in and I tell them I'm injecting, like the demographic, then they turn around and tell me I'm not qualified, and I'm like, 'Well then what the hell are you looking for?'" (Passive recruitment, San Francisco, age ≥30)

Although many participants suggested that social media could reach younger populations and women who would be missed through in-person outreach, others also shared concerns about mistrust of social media. One woman reported that the comments on the recruitment post indicated that people believed that the research opportunity was "sketchy" and did not "look safe". Her prior research experiences helped her evaluate the credibility of the post:

"….[I know] just enough about research to know that this is legit because I've done a lot of research and been part of research studies for other stuff…A lot of people were suspicious even though it was posted by a well-known (community) member, not a researcher." (Social media, Boston, age <30)

Table 4. Key quotations by theme.

| Theme | Representative Quote(s) |
|---|---|
| Participants reported high trust when recruited through community organizations and low trust when recruited through social media. | 1. "…because how can you reach somebody you don't know? To be honest with you, I think that was a good intervention…That would be a good way, if you know somebody. Addicts know addicts. Addicts know that you're not the police."<br>2. "….[I know] just enough about research to know that this is legit because I've done a lot of research and been part of research studies for other stuff…. …. A lot of people were suspicious even though it was posted by a well-known (community) member, not a researcher." |
| Most participants shared that a desire to improve the lives of other women who use drugs drove their motivation to participate in research. | 1. "And I feel like if my whole life, and everything I've done, has saved just one person, just one, then my life has been worth something. And I hope that the things I've told you today, I hope that somewhere down the line it helps somebody."<br>2. "[The outreach staff] said that the research was about getting more help for women in the community…and when she said that, it wasn't about the money for me…I live broke every day right now so I'm used to it, but I want there to be more help for women." |
| Women preferred monetary compensation, which motivated them to participate in the research. | 1. "You have to survive. You have to have food. You got to eat. You got to take care of your habits. And [research participation] is easier than going with some fat, gross pig [to do sex work].." |
| Women participated when research environments were supportive and destigmatized and identified specific risks and barriers that would preclude participation in research. | 1. "Like we were having this conversation, it struck my PTSD post-traumatic stress disorder). And then, I had a very emotional reaction, and it affected me for weeks after. That's not worth any amount of money."<br>2. "Actually, I think a lot of the times women just are so busy dealing with their kids or other life matters that they just don't really want to give it a chance to stop and do something else." |

Despite few participants being recruited through social media, some still recommended it as a strategy to engage with younger women. They suggested sharing information about past research studies to boost credibility and enrolling through online platforms. One participant (social media, Boston, age<30) offered, "maybe links to other research that you've done to prove that you're really researchers and not some weird sting thing."

Overall, the importance of engaging with trusted partners and leveraging past successful research collaborations for both in-person and online recruitment emerged as foundational to build trust with and optimize recruitment of women who use opioids.

Theme 2: Most participants shared that a desire to improve the lives of other women who use drugs drove their motivation to participate in research.

More than half of women reported that participation in research created an opportunity to help other people who use opioids.

"And I feel like if my whole life, and everything I've done, has saved just one person, just one, then my life has been worth something. And I hope that the things I've told you today, I hope that somewhere down the line it helps somebody." (Community partner, San Francisco, age <30)

Another participant highlighted the value of lived experience in shaping research and clinical programs for women who use drugs. As one woman stated, "I think the more input we put in the better help we can get" (Passive recruitment, San Francisco, age ≥ 30).

Although participants reported often feeling ignored by society in general, they recognized that their life experiences offered peers and researchers an avenue to advocate for additional resources, even if it didn't bring them direct benefit:

> "I feel honored to be a part of the research study because maybe by my experience it could help you guys help other women. Or maybe it'll help me if it happens in this lifetime." (Community partner, San Francisco, age ≥30)

In addition to the potential beneficial impacts on others, women suggested that opportunities to share a final product or outcome with research participants would also motivate participation: "Seeing results. Seeing maybe it going somewhere or doing something... Like if what I'm saying is going to affect anybody and help anybody." (passive recruitment, Boston, age < 30)

Even though money was a strong motivator for research participation for many, women said that money alone was not a sufficient motivator. Women were motivated to participate in research when studies were related to helping women who use drugs.

> "[The outreach staff] said that the research was about getting more help for women in the community…and when she said that, it wasn't about the money for me…I live broke every day right now so I'm used to it, but I want there to be more help for women." (Community partner, Boston, age ≥30)

Women wanted their experiences and input to be valued by the researchers. As one participant said, "it's also nice hopefully if someone who you're talking to is not thinking that whatever you say is just stupid and just throws it away like it's trash." (Community partner, Boston, age ≥ 30)

Participants highlighted the importance of being heard, their views being elevated and not dismissed, and the possibility of improving the lives of others as core to the motivation for research engagement.

Theme 3: Women preferred monetary compensation, which motivated them to participate in the research.

Participants identified a range of research study compensation types including food, clothing, cell phones, and transportation passes. However, most participants preferred monetary compensation, specifically cash. Some reported that having money from participating in research mitigated needing to engage in sex work or other activities deemed illegal to generate income:

> "You have to survive. You have to have food. You got to eat. You got to take care of your habits. And [research participation] is easier than going with some fat, gross pig [to do sex work]." (Community partner, Boston, age ≥30)

Almost all participants stated that money was a critical motivator and their preferred method of compensation.

Theme 4: Women participated when research environments were supportive and destigmatized and identified specific risks and barriers that would preclude participation in research.

Although women were enthusiastic about participating in research, they were also aware of the barriers and potential risks of participating. One woman expressed concern about research triggering distressing symptoms that could have an impact beyond the study period:

> "Like we [the research staff and I] were having this conversation, it struck my PTSD (post-traumatic stress disorder). And then, I had a very emotional reaction, and it affected me for weeks after. That's not worth any amount of money." (Snowball sampling, Boston, age ≥30)

Another woman highlighted the potential risks of participating in research, including embarrassment.

> "Embarrassed that somebody's looking at me and in their mind saying, 'Oh, God. This woman's like, you know, whatever.' Like, I'm very self-conscious, so to me I'm thinking somebody's thinking something bad, like, instead of trying to help me they're really like just judging me.." (Community partner, Boston, age ≥30)

In addition to the potential shame and embarrassment that may arise due to disclosures during research participation, women raised logistical barriers including the need for childcare and transportation:

> "Actually, I think a lot of the times women just are so busy dealing with their kids or other life matters that they just don't really want to give it a chance to stop and do something else." (Passive recruitment, San Francisco, age ≥30)

Furthermore, one woman noted that some women may lack autonomy, for example related to intimate partner violence, preventing engagement in research, "He beat me up. We were together for eight years. I had two kids from him… you can't do anything when you're with someone like that. And especially not go to a study." (Community partner, San Francisco, age ≥ 30)

Participants were clear in their need for trauma-informed approaches that consider their past and current experiences, as well as considering the logistics of being a woman (including parenting).

## Discussion

In this qualitative study of 36 women with recent non-prescribed opioid use, we found that community outreach with known organizations and trusted staff was the most effective research recruitment strategy. Many women also expressed hope that research participation could result in better outcomes for other women who use drugs and asked that research findings be shared with research participants. Furthermore, women reported a diversity of barriers and potential risks to research participation. Study designs that use community outreach, that are non-stigmatizing, trauma-informed, and offer equitable compensation could mitigate barriers to research participation for women who use opioids.

Tailoring recruitment to target populations is a critical aspect of study design [20]. This study adds to the existing literature by highlighting specific considerations for researchers who work with women who use opioids. In the setting of continued drug overdose deaths and underrepresentation of women in clinical research, implementing strategies that encourage equitable access to participation are urgently needed and may address potential barriers to women participating in research. For example, partnering with community organizations and individuals who have a strong history of working with women who use drugs to improve enrollment and develop research questions may help ensure that the design is trauma-informed and reflects the values and needs of the community. A 2021 commentary by Simon et al. highlights the need for meaningful research collaborations with people with lived experience and a transition to community-driven research [21]. Many of the barriers to research participation noted by the participants in this study may be minimized if women were partners or leaders in the research process.

Prior research emphasizes the importance of community partners and collaboration in research. The concept of working with community partners for research and the salience of building recruitment and research involvement based on trusted relationships is not novel [22,23]. Many women interviewed expressed that their connection to known organizations involved in recruitment was an important facilitator to research participation and enhanced the credibility of the research team. They expressed a sense of reassurance when community outreach workers knew them well enough to refer them to an appropriate study. Given the history of significant stigmatization of women who use drugs, hesitancy to engage in research is understandable, and recruitment strategies should be developed to minimize stigma and facilitate engagement and trust. Although in San Francisco we did not work directly with community partner organizations, the

highest yield strategy was passive recruitment by organizations that serve people who use drugs. Additionally, a recruitment manager with experience working with relevant community-organizations may have enhanced this strategy.

A recent scoping review found that social media recruitment may be cost effective and useful for reaching populations underrepresented in research [24]. Interestingly, recruitment in our study through social media only led to three participants, despite substantial efforts that included an expert communications and marketing team. In general, our participants highlighted potential challenges related to credibility and the need for prior experience with research to effectively vet the recruitment messaging. We chose social media recruitment in part because we wanted to reach younger women who may not be reached through community outreach. This was validated by one participant who said that social media would be a promising approach to reach women who were not engaged in community-based services. Although in theory it is promising, prior studies using social media platforms to recruit youth have also reported low yield results even when "impressions" or "likes" are high [24]. For example, a study of youth aged 13–20 years in Colorado, recruited through social media, led to contact with 763,613 youth; however, the yield was low, with only 828 youth completing the survey [25] Although we did not design the recruitment images with potential participants, we worked collaboratively with a community partner who serves people who use drugs. Future efforts to use social media could consider collaborating with clients of community partners to develop content. Unexpectedly, snowball sampling through social contacts also did not yield many recruits. Recruitment for this study began two months before the emergence of COVID-19 in the United States and ended in October 2020. During this time, there were rapidly changing messages about physical distancing (often called social distancing at the time) and significant changes in how social networks interacted with each other, leading to a decreased number of contacts [26] It is likely that these shifts had an impact on the low yield of snowball sampling.

Women in this study highlighted the importance of providing the social and emotional supports to research participants in addition to compensation to recruit women. They noted that research practices may trigger past trauma and exacerbate experiences of stigma if research designs (and training of research team members) do not address this. Women's input in the research process is critical to ensure the content of an interview guide or intervention is not harmful and avoids triggering trauma symptoms. Researchers should also apply some general approaches to conducting research with marginalized populations to ensure their safety [27] Although not specifically noted by the participants, taking specific steps such as adequately preparing participants for the interview content, offering breaks during the interview, and having resources available in case the research triggers trauma or mental health issues can help create a safe environment for participants. In addition, staff should be trained in trauma-informed approaches including being prepared to respond to distressed participants and share appropriate resources or supports [28]. For some women, child caretaking responsibilities may be a barrier. Researchers could consider providing childcare or depending on the interview content and age of child allow them to be present, conducting study visits outside of regular business hours, or providing alternative participation formats such as phone interviews. Finally, providing participants with clear information about how the research team will ensure protection of their confidentiality and safety is critical.

The importance and preference of monetary compensation for research participants has been described in other populations [29,30]. Concerns regarding cash or card payments being used to purchase drugs are paternalistic, as identified by participants in our study and other reviews assessing compensating approaches for people who use drugs. Cash payments were the preferred method of compensation and may be a more equitable and effective recruitment incentive than card or other forms of monetary payments for women who use opioids. Fair compensation for participation is ethical and expected for other diseases and disorders, and the same should be expected for research among people who use substances.

Trust and alignment of study goals with improving the lives of women who use drugs was also critical to engaging women. Women were clear in their desire to improve the lives of others, particularly other women who use drugs. Without input from women with lived experience, there is a risk of asking research questions or choosing outcomes that are not relevant or that are triggering or harmful. Currently, much substance use research includes outcomes such as drug

use frequency measures and urine drug testing; however, our participants were interested in research that improved the quality of life for women who use drugs. Using community-engaged methods, women can be partners in designing studies and identifying outcomes that are most relevant to them and their communities. Women highlighted additional steps that researchers could take such as sharing the results of research studies with participants at the end of a study, including them in the interpretation of results, and collaborating to disseminate the results among the community. Without the input of women, there is a significant risk of their continued under-representation in research, a lack of gender-responsive studies, and further marginalization and isolation of knowledge generated in substance use research. Although not a comprehensive list, researchers could consider asking the following questions prior to engaging in research with populations that have been historically excluded from research participation.

1. What community partners or others with lived experience have contributed to the design of the study?

2. What trauma-informed approaches have been integrated into study design, including training of the research team about the population being studied?

3. How has compensation for participation been determined and is it fair and equitable?

4. How will results be shared back with the research participants?

5. Can research participants collaborate to disseminate the results?

## Limitations

We describe several limitations in our study. Early in our study, public health policies to minimize the spread of COVID-19 were enacted, limiting our ability to recruit for in-person interviews and engage directly with community partners. Prior to COVID-19, we completed much of the enrollment though community outreach in Boston; however, we cannot ascertain the extent to which snowball sampling methods, which rely on person-to-person interactions, were impacted by this necessary adaptation in recruitment technique. Thus, snowball sampling may have been more efficient in other circumstances.

Recruiting women under the age of 30 remained a challenge. Social media was specifically mentioned by some participants as a strategy to reach younger women; however, given the potential for distrust, credible messages are crucial. Though we developed recruitment materials, including social media ads with a community partner, young women were not involved in the development of the study or recruitment messaging. Our difficulty recruiting young women highlights the need to build trusting partnerships with organizations that serve young women who can help to create meaningful research that is responsive to them. Possible strategies to improve youth recruitment could include partnering with youth-specific organizations, using text messaging or other youth-informed messaging, and including youth in the design of the study.

## Conclusions

In an exploratory study of women who use opioids, dedicated recruitment resources through trusted community outreach recruited the most women and women described recruitment through known networks to be the most effective approach. Although this study was conducted in two urban areas, findings could be applicable to rural or resource-limited settings. For example, the principles of being trauma-informed, providing fair compensation, and engaging with community partners and people with lived or living experience are not dependent on urbanicity. Throughout the interviews, women expressed a desire for their voices to be valued through the sharing of results, a preference for monetary compensation, and a hope that they could improve outcomes for women who use drugs. Participants also preferred monetary compensation for research participation. To improve enrollment of diverse and representative samples in research, investigators should

carefully consider the population of interest and collaborate with people with lived experience and community organizations to ensure that study designs and recruitment methods minimize potential harms and optimize research acceptability and participation.

## Acknowledgments

We would like to thank the Boston Medical Center Communications and Marketing team for their support in our social media recruitment.

## Author contributions

**Conceptualization:** Sarah M. Bagley, Ariel D. Maschke, Alexander Y. Walley, Phillip O. Coffin, Christine M. Gunn.

**Data curation:** Ariel D. Maschke, John Farley.

**Formal analysis:** Sarah M. Bagley, Ariel D. Maschke, Miriam T.H. Harris, Vanessa M. McMahan, Cynthia Barrett, Christine M. Gunn.

**Methodology:** Sarah M. Bagley, Ariel D. Maschke, Miriam T.H. Harris, Alexander Y. Walley, Phillip O. Coffin, Christine M. Gunn.

**Project administration:** Sarah M. Bagley, Christine M. Gunn.

**Writing – original draft:** Sarah M. Bagley, Alexander Y. Walley, Samantha Johnson, Emily Hurstak.

**Writing – review & editing:** Sarah M. Bagley, Ariel D. Maschke, Miriam T.H. Harris, Alexander Y. Walley, Samantha Johnson, Emily Hurstak, John Farley, Sarah G. Keller, Vanessa M. McMahan, Cynthia Barrett, Phillip O. Coffin, Christine M. Gunn.

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
