## [Decision Letter · Decision Letter 0]

25 Jun 2025

Dear Dr. Bagley, 

Thank you for submitting your manuscript to PLOS ONE. After careful consideration, we feel that it has merit but does not fully meet PLOS ONE’s publication criteria as it currently stands. Therefore, we invite you to submit a revised version of the manuscript that addresses the points raised during the review process.

We look forward to receiving your revised manuscript.

Kind regards,

Ali Ahmed, PhD

Academic Editor

PLOS ONE

Journal Requirements:

“This work was supported by the National Institutes of Health [R01DA045690].”

6. We note that Traditional & Meme Images includes an image of a participant in the study. 

As per the PLOS ONE policy (http://journals.plos.org/plosone/s/submission-guidelines#loc-human-subjects-research) on papers that include identifying, or potentially identifying, information, the individual(s) or parent(s)/guardian(s) must be informed of the terms of the PLOS open-access (CC-BY) license and provide specific permission for publication of these details under the terms of this license. 

Please download the Consent Form for Publication in a PLOS Journal (http://journals.plos.org/plosone/s/file?id=8ce6/plos-consent-form-english.pdf). The signed consent form should not be submitted with the manuscript, but should be securely filed in the individual's case notes. 

Please amend the methods section and ethics statement of the manuscript to explicitly state that the patient/participant has provided consent for publication: “The individual in this manuscript has given written informed consent (as outlined in PLOS consent form) to publish these case details”. 

Reviewers' comments:

Reviewer's Responses to Questions

**Comments to the Author**

1. Is the manuscript technically sound, and do the data support the conclusions?

Reviewer #1: Yes

Reviewer #2: Yes

2. Has the statistical analysis been performed appropriately and rigorously?

Reviewer #1: N/A

Reviewer #2: N/A

3. Have the authors made all data underlying the findings in their manuscript fully available?

Reviewer #1: No

Reviewer #2: Yes

4. Is the manuscript presented in an intelligible fashion and written in standard English?

Reviewer #1: Yes

Reviewer #2: Yes

Reviewer #1: 1. This is a compelling and methodologically sound qualitative study exploring the motivations and barriers for research participation among women who use opioids—an underserved and often misrepresented group in research.

2. Could you please clarify the qualitative study design used? While the study appears to follow a general qualitative approach with thematic analysis, it would be helpful to explicitly state whether it aligns with phenomenology, grounded theory, or another framework. Naming the design strengthens methodological transparency.

3. The sample size of 36 participants is appropriate for the aims of the study, but it would be useful to elaborate more on how thematic saturation was assessed. Was it based on real-time monitoring during interviews or retrospectively after coding?

4. The use of both deductive and inductive coding is well-justified. Including a brief note on how reflexivity was maintained or how researcher bias was managed would enhance credibility.

5. Theme presentation was rich and supported with strong participant quotations. However, the narrative format can get dense. I suggest adding a figure or visual summary (e.g., a thematic map, diagram, or table) that clearly shows the four main themes and their sub-themes or illustrative quotes. This could help readers better visualize relationships among themes and digest findings more easily.

6. Recruitment findings were detailed and informative, particularly the comparison across different strategies. It might be useful to include a short synthesis paragraph in the results that compares these strategies side-by-side, ideally with a visual (e.g., bar chart or flow diagram).

7. The adaptation of recruitment strategies due to COVID-19 was well-described. If possible, briefly note whether virtual methods (e.g., phone interviews) influenced participants’ openness or depth of disclosure.

8. Snowball sampling had low yield. Aside from pandemic disruptions, were there interpersonal or cultural factors at play? Was compensation sufficient or was trust a barrier?

9. The strong emphasis on trust, stigma, and trauma-informed practices is commendable. You might want to include a short framework or checklist that other researchers could adopt when working with similar populations.

10. The quotes were powerful, particularly those relating to compensation and feeling heard. That said, some could be trimmed for clarity while still retaining impact.

11. The paper does a great job showing how monetary compensation was not just a motivator, but also an ethical and dignity-affirming issue. This is a valuable contribution to current debates on incentives in research with vulnerable groups.

12. The underrepresentation of younger women is noted in the limitations, but suggestions for overcoming this (e.g., youth-informed messaging, peer navigators) would be helpful additions.

13. There’s strong potential to expand community-engaged practices in future research. Was there any participant or peer involvement in tool development or analysis? If not, it could be something to explore moving forward.

14. The findings are clearly situated in the urban U.S. context. A sentence or two on how findings might transfer to rural or resource-limited settings would help with broader applicability.

15. Writing is generally clear, but transitions between themes in the results could be refined. Consider adding a summary sentence at the end of each theme section to reinforce key takeaways.

16. The tables are useful, but for clarity, consider highlighting or color-coding cells that show the most effective recruitment strategies or participant characteristics that align with each strategy.

Reviewer #2: Interesting, coherently written and potentially academically useful paper highlighting possible facilitators of recruitment to research studies of research participators from women who use non-prescribed opioids.

**Do you want your identity to be public for this peer review?** For information about this choice, including consent withdrawal, please see our Privacy Policy

Reviewer #1: No

Reviewer #2: No

---

## [Author Response · Author response to Decision Letter 1]

13 Oct 2025

Date August 8, 2025

Dear Dr. Ahmed :

We are grateful for the careful review of our manuscript, “¬¬¬¬¬¬¬¬¬¬¬¬¬¬¬¬A Qualitative Study Exploring Motivations for Participating in Research among Women who Use Opioids”. We believe that reviewer comments have strengthened our work and appreciate your consideration of this manuscript for publication. As requested, we have included the following sentence in the cover letter, "The funders had no role in study design, data collection and analysis, decision to publish, or preparation of the manuscript."

Below are detailed responses to overall and reviewer comments. Per revision instructions, we have also included two versions of the manuscript, one with tracked changes and one clean copy in this re-submission. We look forward to answering any further questions or responding to further comments that you or the reviewers may have.

Sincerely,

Sarah M. Bagley, MD, MSc

Associate Professor of Medicine and Pediatrics

Director of the Youth Advocacy and Research Collaborative

Chobanian & Avedisian School of Medicine and Boston Medical Center

Overall comments

Comment: Please ensure that your manuscript meets PLOS ONE's style requirements, including those for file naming. The PLOS ONE style templates can be found at

Response: We have ensured that manuscript meets PLOS ONE's style requirements, including those for file naming.

Comment: We note that the grant information you provided in the ‘Funding Information’ and ‘Financial Disclosure’ sections do not match.

Response: Thank you for pointing this out, we have ensured that funding information matches.

Comment: Thank you for stating the following financial disclosure:

“This work was supported by the National Institutes of Health [R01DA045690].”

Response: We have added the sentence to the cover letter.

Comment: We note that you have indicated that there are restrictions to data sharing for this study. For studies involving human research participant data or other sensitive data, we encourage authors to share de-identified or anonymized data. However, when data cannot be publicly shared for ethical reasons, we allow authors to make their data sets available upon request. For information on unacceptable data access restrictions, please see http://journals.plos.org/plosone/s/data-availability#loc-unacceptable-data-access-restrictions.

Response: We very much recognize the importance of data sharing and transparency. We also acknowledge that participant data does include sensitive topics such as use of illegal substances and sex work. When we were collecting informed consent in 2020, we did not include disclosures about data sharing that would allow participants to indicate their consent for researchers to share data with others outside the research team. It is our institutional practice to include a statement about the potential for data sharing to ensure consent for that process, we therefore do not think that it is appropriate to deposit these data in a repository.

Comment: Please include captions for your Supporting Information files at the end of your manuscript, and update any in-text citations to match accordingly. Please see our Supporting Information guidelines for more information: http://journals.plos.org/plosone/s/supporting-information.

Response: We have included captions for our Supporting Information files at the end of the manuscript.

Comment: We note that Traditional & Meme Images includes an image of a participant in the study.

Response: We appreciate the concern of including a participant image and can confirm that the Meme Image is from a stock photo and is not a participant in the study. In the text of the appendix describing the development of the Social Media Campaign, we have the following sentence “The Meme Image is from a stock photo and is not a participant in the study.” If the editor would like us to provide that information in another setting or more prominently, please let us know.

Reviewer #1

Comment: This is a compelling and methodologically sound qualitative study exploring the motivations and barriers for research participation among women who use opioids—an underserved and often misrepresented group in research.

Response: We appreciate the reviewer’s comments on our study.

Comment: Could you please clarify the qualitative study design used? While the study appears to follow a general qualitative approach with thematic analysis, it would be helpful to explicitly state whether it aligns with phenomenology, grounded theory, or another framework. Naming the design strengthens methodological transparency.

Response: We used a directed qualitative content analysis and added a citation in the methods section to make this clearer.

Comment: The sample size of 36 participants is appropriate for the aims of the study, but it would be useful to elaborate more on how thematic saturation was assessed. Was it based on real-time monitoring during interviews or retrospectively after coding?

Response: Our study team met weekly during recruitment and while interviews were being conducted. We discussed what concepts were emerging and assessed when thematic saturation was being approached and then reached. As noted in the limitations, we did have a difficult time recruiting younger participants and it is possible that had we been more successful in recruiting them, additional themes would have emerged. As we had otherwise reached thematic saturation, we decided to stop enrollment in the study and include the low recruitment of young women as a limitation.

Comment: The use of both deductive and inductive coding is well-justified. Including a brief note on how reflexivity was maintained or how researcher bias was managed would enhance credibility.

Response: Thank you for this comment, we have added the following comment to the methods:

“The core coding team met weekly to discuss emerging themes, consensus, and when coding was completed the interpretation of results. The team included individuals with diverse experience and expertise in community engagement, risk communication, addiction medicine and harm reduction, and women’s health. In weekly meetings, the team held robust conversations about data to mitigate potential researcher bias.”

Comment: Theme presentation was rich and supported with strong participant quotations. However, the narrative format can get dense. I suggest adding a figure or visual summary (e.g., a thematic map, diagram, or table) that clearly shows the four main themes and their sub-themes or illustrative quotes. This could help readers better visualize relationships among themes and digest findings more easily.

Response: This a great suggestion, we have added a Table 4 that highlight the themes and the representative quotes.

Comment: Recruitment findings were detailed and informative, particularly the comparison across different strategies. It might be useful to include a short synthesis paragraph in the results that compares these strategies side-by-side, ideally with a visual (e.g., bar chart or flow diagram).

Response: We are grateful that our recruitment findings were informative. In the second paragraph of the Results and in Table 2 we describe the yield of each of the recruitment strategies.

Comment: The adaptation of recruitment strategies due to COVID-19 was well-described. If possible, briefly note whether virtual methods (e.g., phone interviews) influenced participants’ openness or depth of disclosure.

Response: This is an excellent question. Unfortunately, this is not information that we collected so we are not able to report on how virtual methods impacted openness or depth of disclosure.

Comment: Snowball sampling had low yield. Aside from pandemic disruptions, were there interpersonal or cultural factors at play? Was compensation sufficient or was trust a barrier?

Response: This is an important point, and it surprised us that this recruitment method yielded such low results. We believe that the pandemic was the major driver and did not detect interpersonal or cultural factors at play. In both recruitment sites, we had strong relationships with community partners, and compensation was like other projects, and thus we do not think that was an issue.

Comment: The strong emphasis on trust, stigma, and trauma-informed practices is commendable. You might want to include a short framework or checklist that other researchers could adopt when working with similar populations.

Response: Thank you for noting the importance of these principles. We have included a short checklist in the Discussion that includes the following questions for researchers to consider when working with similar populations.

“Although not a comprehensive list, researchers could consider asking the following questions prior to engaging in research with populations that have been historically excluded from research participation.

1. What community partners or others with lived experience have contributed to the design of the study?

2. What trauma-informed approaches have been integrated into study design, including training of the research team about the population being studied?

3. How has compensation for participation been determined and is it fair and equitable?

4. How will results be shared back with the research participants?

5. Can research participants collaborate to disseminate the results?”

Comment: The quotes were powerful, particularly those relating to compensation and feeling heard. That said, some could be trimmed for clarity while still retaining impact.

Response: We agree that the quotes were powerful and are grateful to the research participants for sharing their perspectives. We have trimmed the quotes for clarity while retaining impact

Comment: The paper does a great job showing how monetary compensation was not just a motivator, but also an ethical and dignity-affirming issue. This is a valuable contribution to current debates on incentives in research with vulnerable groups.

Response: We appreciate the reviewer’s perspective and agree that elevating the issue of fair compensation is a critical consideration when engaging in research with vulnerable groups.

Comment: The underrepresentation of younger women is noted in the limitations, but suggestions for overcoming this (e.g., youth-informed messaging, peer navigators) would be helpful additions.

Response: As the reviewer noted, the underrepresentation of younger women was a limitation of this study. As suggested, we have added the following sentence to the limitations to suggest possible strategies to improve their involvement in future studies.

“Possible strategies to improve youth recruitment could include partnering with youth-specific organizations, using text messaging or other youth-informed messaging, and including youth in the design of the study.”

Comment: There’s strong potential to expand community-engaged practices in future research. Was there any participant or peer involvement in tool development or analysis? If not, it could be something to explore moving forward.

Response: We agree with this idea of including participants and peers in interview guide development and analysis in the future, although this was not our approach in this study.

Comment: The findings are clearly situated in the urban U.S. context. A sentence or two on how findings might transfer to rural or resource-limited settings would help with broader applicability.

Response: This is a great point, we have the following to the discussion.

“Although this study was conducted in two urban areas, findings could be applicable to rural or resource-limited settings. For example, the principles of being trauma-informed, providing fair compensation, and engaging with community partners and people with lived or living experience are not dependent on urbanicity.”

Comment: Writing is generally clear, but transitions between themes in the results could be refined. Consider adding a summary sentence at the end of each theme section to reinforce key takeaways.

Response: We have added summary sentences at the end of each theme section to reinforce key takeaways as suggested by the reviewer.

Comment: The tables are useful, but for clarity, consider highlighting or color-coding cells that show the most effective recruitment strategies or participant characteristics that align with each strategy.

Response: We appreciate that it would be helpful to be able to distinguish the differences between recruitment strategies and participant characteristics in the tables. We attempted to use highlighting or shading but found that the tables were too busy. If the reviewer has specific suggestions about how to accomplish this, we would be happy to make the edits.

Reviewer #2:

Comment: Interesting, coherently written and potentially academically useful paper highlighting possible facilitators of recruitment to research studies of research participators from women who use non-prescribed opioids.

Response: We appreciate Reviewer 2’s assessment of our manuscript and hope that it would improve recruitment and participation of women who use non-prescribed opioids in research.

---

## [Editor Report · Decision Letter 1]

20 Oct 2025

A qualitative study exploring motivations for participating in research among women who use opioids

PONE-D-24-44146R1

Dear Dr. Sarah,

We’re pleased to inform you that your manuscript has been judged scientifically suitable for publication and will be formally accepted for publication once it meets all outstanding technical requirements.

Kind regards,

Ali Ahmed, PhD

Academic Editor

PLOS ONE
---

## [Editor Report · Acceptance letter]

PONE-D-24-44146R1

PLOS ONE

Dear Dr. Bagley,

I'm pleased to inform you that your manuscript has been deemed suitable for publication in PLOS ONE. Congratulations! Your manuscript is now being handed over to our production team.

Kind regards,

on behalf of

Dr. Ali Ahmed

Academic Editor

PLOS ONE